# A Vibration Sensing Device Using a Six-Axis IMU and an Optimized Beam Structure for Activity Monitoring

**DOI:** 10.3390/s23198045

**Published:** 2023-09-23

**Authors:** Pieter Try, Marion Gebhard

**Affiliations:** Group of Sensors and Actuators, Department of Electrical Engineering and Applied Sciences, Westphalian University of Applied Sciences, 45897 Gelsenkirchen, Germany; marion.gebhard@w-hs.de

**Keywords:** inertial measurement unit, sensor fusion, signal processing, structural vibration measurement, activity estimation

## Abstract

Activity monitoring of living creatures based on the structural vibration of ambient objects is a promising method. For vibration measurement, multi-axial inertial measurement units (IMUs) offer a high sampling rate and a small size compared to geophones, but have higher intrinsic noise. This work proposes a sensing device that combines a single six-axis IMU with a beam structure to enable measurement of small vibrations. The beam structure is integrated into the PCB of the sensing device and connects the IMU to the ambient object. The beam is designed with finite element method (FEM) and optimized to maximize the vibration amplitude. Furthermore, the beam oscillation creates simultaneous translation and rotation of the IMU, which is measured with its accelerometers and gyroscopes. On this basis, a novel sensor fusion algorithm is presented that adaptively combines IMU data in the wavelet domain to reduce intrinsic sensor noise. In experimental evaluation, the proposed sensing device using a beam structure achieves a 6.2-times-higher vibration amplitude and an increase in signal energy of 480% when compared to a directly mounted IMU without a beam. The sensor fusion algorithm provides a noise reduction of 5.6% by fusing accelerometer and gyroscope data at 103 Hz.

## 1. Introduction

The measurement and analysis of structural vibrations has a wide range of applications. Traditionally, it is used in predictive maintenance to detect the failure of rotating machines [1,2,3]. The constant vibration of rotating machinery and the high correlation of failure modes and vibration patterns make this application very useful and particularly widespread. Vibration monitoring is also used in structural health monitoring, where vibration sensors are used to assess the condition of buildings and civil structures [4,5]. A key application is monitoring of the occurrence, formation and propagation of damage.

Another application is demonstrated in recent studies, where structural vibration of ambient objects is used to detect and estimate the activity of living creatures. These methods are able to obtain a wide range of information such as human occupancy [6], foot step location [7], different human activities [8,9] as well as animal activity of pigs [10] and mice [11]. When living creatures perform physical activities, they interact with ambient objects such as the floor, walls and furniture. These interactions cause the ambient objects to vibrate. The vibration contains different frequency components that are related to the oscillation modes and are unique for different interactions. The main challenges are the measurement of small vibrations and the data analysis to detect and classify activity. In most instances, geophones are used to measure vibrations because they provide low noise due to their large proof mass. However, geophones have several weaknesses that make them unsuitable in some scenarios.

In our previous work, non-contact activity monitoring of mice is demonstrated, which uses a commercially available multi-axial inertial measurement unit (IMU) to measure activity-induced structural vibrations of the husbandry cage [11]. This method enables activity monitoring in large-scale mouse husbandry, where thousands of mice are held in individually ventilated husbandry cages that are stacked in racks. There are established methods for activity monitoring of mice which are not suitable for large-scale husbandry due to cost. These use radar [12], cameras [13,14] or electromagnetic detection methods [15,16]. A method using low-cost sensors enables large-scale monitoring, reducing the workload for human caretakers, improving response time for animal care and providing continuous long-term activity data for studies. In our previous work, the IMU is mounted in the center on the outside of the cage floor and continuously monitors cage vibrations in three spatial directions. These data are split into 10 s intervals and preprocessed, then a classification algorithm is used to estimate the activity level of the mouse inside. In this scenario, geophones are unsuitable, because they cannot physically fit below a cage due to the cages being stacked in racks. A large geophone would also interfere in the handling of the cages. Furthermore, the husbandry cage is observed to exhibit vibrations with a frequency of over 250 Hz, which is above the frequency range of commonly used geophones [17]. In contrast, IMUs provide many benefits in this use case, including small size, low weight, high performance-to-cost ratio, high sampling rate and multiple sensors in a single chip. Especially, the multiple sensors are shown to provide additional information that improves classification accuracy. However, a disadvantage of commercial micro-electro-mechanical systems (MEMS) sensors is their intrinsic noise. As a result, very small vibrations cannot be detected, thus impacting the classification accuracy. A promising technique used in several applications is the functional use of the supporting structure e.g., efficient energy harvest [18]. This approach is also shown to improve state estimates in robotics by taking the flexibility of the robotic limbs into account [19].

In this work, a sensing device is proposed that uses a mechanical beam structure to achieve improved vibration sensing with a single multi-axial IMU. The beam is integrated in the printed circuit board (PCB) of the device, where the IMU is mounted on one end of the beam and the other end is mounted to the ambient object. This arrangement significantly increases the vibration amplitude at the IMU location, thereby increasing the signal-to-noise ratio (SNR). Furthermore, a sensor fusion algorithm is presented that combines the data from the accelerometers and gyroscopes to reduce the intrinsic noise of the sensor. The sensor fusion is based on the correlation of the linear acceleration and angular rotation of the IMU, which occurs due to certain oscillation modes. The key contributions of this work are as follows:A novel method is presented to enhance the vibration sensing capabilities of multi-axial sensors, such as IMUs, and the concept is evaluated in simulation and experiment.A PCB-based vibration amplifying beam structure is designed, optimized and experimentally validated.A novel sensor fusion algorithm is presented, which uses the unique oscillation modes of the beam structure to combine multiple sensor signals to reduce noise. The algorithm is suitable for arbitrary multi-axis sensors and beam structures.An experimental study is devised to evaluate the concept in an application environment, where the desired vibration amplification and noise reduction through sensor fusion is successfully demonstrated.

## 2. Materials and Methods

In previous work [11] it is observed that many physical activities only induce very small vibrations $z_{1}$, which are below the noise floor of commercial IMUs. The proposed sensing device aims to amplify these vibrations to improve the detection of activities that only generate very small vibrations. Section 2.1 and Section 2.2 outline the generalized model for the oscillation of ambient objects. Section 2.3 and Section 2.4 illustrate the FEM simulation to analyze the use case and generate an optimized beam design. In Section 2.5, the sensor fusion algorithm is presented, which combines accelerometer and gyroscope data to reduce the intrinsic sensor noise. The experimental setup is shown in Section 2.6, which serves to evaluate the proposed concept.

### 2.1. Physical Model of Activity-Induced Vibration

Previous work has shown that living creatures create structural vibrations through different force pulses during physical activity. These forces cause the vibration and natural oscillation of ambient objects. A generalized physical model of an environmental object is given by a spring mass damper system. The forces that are generated by physical activity act on the mass and cause oscillation. This model is illustrated in (A) of Figure 1.

In the regular setup labeled as (A), the sensor is directly mounted to the object. The environmental object usually has a mass *m* that is much greater than the mass of the IMU $m_{IMU}$. Thus, the influence of the IMU can be neglected. Physical activity of living beings applies force pulses $P=\int F_{Activity}dt$ on the object, which generates dampened natural oscillations. Given the rigid mounting of the sensor, the position of the sensor $z_{2}$ is equal to the position of the environmental object $z_{1}$.

### 2.2. Beam Structure for Vibration Amplification

The proposed sensing device employs an additional spring mass damper system situated between the environmental object and the IMU, which is shown in (B) of Figure 1. By optimizing the parameters $m_{IMU}$, $k_{2}$ and $d_{2}$ of this spring system, an amplified oscillation $z_{2}>z_{1}$ can be achieved. This setup forms a two-mass-spring damper system. Depending on the scenario, the influence of the sensing device on the environmental object cannot be neglected. Analytical analysis of systems with complex geometry is near impossible and numerical methods are generally used to find solutions. In this work, finite element method (FEM) analysis in Ansys Mechanical [20] is employed to analyze the oscillating system, which is presented in Section 2.3 and Section 2.4.

It is proposed to implement the additional spring mass damper system as a beam structure. One end of the beam is mounted to the ambient object and the IMU is mounted on the other end. Furthermore, the beam is integrated in the PCB of the device and contains the electrical connections. This approach has many benefits, such as ease of manufacturing and implementation in devices. The beam is designed and optimized using FEM simulation, which is described in detail in Section 2.4. Although not suitable for determining vibration characteristics, the classical equations for the deflection of a beam under load [21] are helpful in formulating design criteria. The forces that act on the sensing device are shown in Figure 2.

An activity-induced force pulse $P=\int F_{Activity}dt$ acts vertically on the cage floor, causing a natural oscillation of the cage. As described in Section 2.3, the cage floor oscillates like a membrane. The displacement $z_{1}$ of the cage floor oscillation exerts a force on the sensing device. This force acts as a vertical reaction force $F_{RV}$ on the base of the beam. It is directly opposed by the inertial force of the beam $W_{Beam}$ and sensor $F_{IMU}$. In order to calculate the displacement at the end of the beam, where the IMU is located, $F_{IMU}$ is neglected as it is much smaller than $W_{Beam}l$. The tilt angle and displacement are calculated as:(1)$$\theta =\frac{W_{Beam}l^{3}}{6EI_{y}},$$
(2)$$z_{2}=\frac{W_{Beam}l^{4}}{8EI_{y}}=\theta \frac{4l}{3},$$
where $z_{2}$ is the displacement, θ is the tilt angle, *E* is the Young’s modulus and $I_{y}$ is the moment of inertia [21]. Equation (Equation 2) further shows that the displacement $z_{2}$ and the tilt angle θ are linearly correlated. As a six-axis IMU is capable of measuring both variables directly, this circumstance allows for sensor fusion, which is further described in Section 2.5.

For proof of concept, a first design of the beam is proposed here. Other designs will be explored in the future. This design resembles a bending beam, which is split laterally and folded to form two parallel braces that are shaped around the base. This design shifts the center of mass further to the base of the beam, which reduces the moment generated by the beam. It is shown in Figure 3 and Figure 4. The manufacturing of the prototype requires some design restrictions: The beam needs a surface to mount to the cage and one to mount the IMU, where the surfaces have a size of 17 mm × 17 mm each. The braces need a minimum width *w* of 3 mm to accommodate the data lines. The interspace in the beam needs to be at least 4 mm wide.

For prototyping purposes, the system on a chip (SoC) of the IMU is not mounted directly to the beam structure but to a small daughter board shown in Figure 4, which contains the IMU. This daughter board is easily desoldered and mounted on another prototype for testing without damaging the delicate SoC.

### 2.3. FEM Analysis of the Ambient Object

The design of the beam structure needs to be adapted to the vibration characteristics of the object. The ambient object in this use case is the husbandry cage HRC500 from Zoonlab, which is evaluated in the finite element analysis (FEA) software Ansys Mechanical. Some simulation results are illustrated in Figure 5. The simulation uses a CAD model of the HRC500 cage from Zoonlab [22]. The parameters for the PSU material of the cage were adopted from the Ansys library with a change to the density to match the real weight of the cage. The cage is fixed on its lateral side rails, which are on either side of the cage. In addition, a distributed mass of 200 g is applied to the bottom of the cage to simulate the wood flocking that typically covers the floor.

Figure 5 illustrates the first four oscillation modes of the Zoonlab HRC500 husbandry cage. The 1st mode shows a lateral displacement of the entire cage. The 2nd to 4th modes are membrane-like oscillations of the cage floor with different numbers of peaks and nodes. The 2nd mode has one peak and no nodes. The 3rd mode has two peaks and one node, while the 4th mode has three peaks and two nodes. The simulation predicts many more oscillation modes that follow a similar pattern.

The simulation results suggest that the cage primarily exhibits a membrane-like oscillation of the cage floor. This is also observed in previous work [11]. Furthermore, when the cage floor is excited by a vertical force pulse, the 2nd mode has the highest displacement. It is therefore concluded that the center of the cage floor is the ideal location for the IMU.

### 2.4. FEM Analysis Beam Design

In order to optimize the design shown in Figure 3, a large number of parameters and dimensions can be adjusted. Some parameters are already set due to manufacturing, assembly and mounting requirements. The remaining parameters are the length, height, width, Young’s modulus and mass. While the mass is proportional to the overall size, the height and Young’s modulus can only be changed by using a different raw material. Therefore, the main parameters for optimizing the design are the length and width. These are defined in Figure 3 as the width *w* of each of the symmetrical braces and the length *l* of the outer brace.

In order to analyze the beam structure in its intended use case scenario, the simulation includes the cage as described in Section 2.3. The beam structure is mounted in the middle of the cage floor with a spacer to allow for oscillation. The geometry of the sensor board is not modeled since it does not influence the oscillation characteristics, but the mass of the sensor board is simulated with a surface mass. The IMU is represented by a 2 mm × 3 mm × 2 mm rectangle and assembled on the sensor board mount surface. The material characteristics of the beam are set to a standard FR-4 glass-reinforced epoxy laminate material from the Ansys library.

The modal analysis depicts the oscillation characteristics of the beam when mounted on the cage and shows that the beam exhibits many oscillation modes. However, some of them have a similar form of deformation and typically occur in the same frequency range. The oscillation modes are grouped together based on their deformation form. In Figure 6 four different deformation forms are presented of a beam structure with w=3 mm and l=29 mm.

The first deformation form appears in the range of 99 Hz to 118 Hz and exhibits a bend in the *z*-axis, which is accompanied by a rotation around the *x*-axis. The second form shows a sideways movement in the *x*-axis and occurs between 398 Hz to 426 Hz. The third form shows a twisting deformation around the *y*-axis and appears in the range of 648 Hz to 745 Hz. The fourth form shows a twisting motion around the *x*-axis and a back-and-forth movement in the y-direction. It occurs in the range of 878 Hz to 1101 Hz. Beam structures with similar dimensions exhibit the same deformation forms at different frequencies.

A transient structural FEM analysis is performed to evaluate the activity-induced structural vibration of the cage. The simulation is 0.3 s long with a step time of 50 μs. At 1 ms, a force pulse of P=0.15N×1ms=150μNs is applied to the center of the cage floor. Afterwards, the system continues its dampened oscillation undisturbed. The result of the simulation is the acceleration at the location of the virtual IMU in the *x*-, *y*- and *z*-direction. The amplitude of the oscillation modes is evaluated in the frequency domain by Fourier transforming the output acceleration data.

Based on this simulation setup, a design with maximum vibration magnitude is identified in Section 3.1 by comparing beams with different *w* and *l* dimensions to determine optimal dimensions.

### 2.5. Sensor Fusion Algorithm

The IMU data consist of six signals, which are the three acceleration signals $a_{x}$, $a_{y}$ and $a_{z}$ and the three angular rate signals $\omega_{x}$, $\omega_{y}$ and $\omega_{z}$. In the standard configuration, where the sensor is mounted directly on an object, the different sensor signals of the IMU are not directly correlated and cannot be fused in the time domain e.g., to reduce noise. This is shown in [11] where statistical information from different sensors is combined to improve classification accuracy. However, the signals could not be combined on a per-sample basis to reduce noise or increase the SNR.

The simulation of the beam structure in Section 2.4 shows that certain oscillation modes of the beam are composed of displacement in multiple axes. This creates a mechanical coupling between the different signals of the IMU, which enables sensor fusion. Using prior knowledge of the oscillation modes, a sensor fusion algorithm is proposed that combines multiple correlated sensor signals to create a single signal with lower noise. As the SNR varies between signals, a main signal is determined first, which has the highest SNR. The correlated signals are then combined to reduce the noise of the main signal.

The sensor signals are modeled as a vibration *f* with additive Gaussian noise with zero mean. The noise of the different signals is uncorrelated, since the intrinsic sensor noise is mainly due to the thermal noise of the proof mass in each sensor. The underlying theory is presented in the following. Two signals of different sensors are denoted as:(3)$$x=f+N_{x}\left(\mu =0,\sigma_{x}^{2}\right),$$
(4)$$y=f+N_{y}\left(\mu =0,\sigma_{y}^{2}\right),$$
where *x* and *y* are the two measurements, N denotes the corresponding additive noise, $\sigma^{2}$ is the variance of noise and *f* is the vibration. A weighted estimator *z* is constructed to combine the signals *x* and *y*:(5)z=cx+(1−c)y,
(6)$$z=f+N_{z}\left(\mu =0,\sigma_{z}^{2}\right),$$
(7)$$\sigma_{z}^{2}=c^{2}\sigma_{x}^{2}+{\left(1-c\right)}^{2}\sigma_{y}^{2},$$
where $N_{z}$ is the noise of the estimator *z*, $\sigma_{z}$ is the noise variance and *c* is the weight. By minimizing $\sigma_{z}$ in Equation (Equation 7), the optimal value of *c* is determined as:(8)$$c=\frac{\sigma_{x}^{-2}}{\sigma_{x}^{-2}+\sigma_{y}^{-2}}$$

In the ideal case $\sigma_{x}=\sigma_{y}$, the weight is c=0.5. Using Equation (Equation 7), the standard deviation of noise of the estimator is calculated as $\sigma_{z}\approx 0.707\sigma_{x}$. For an example with two signals with $\sigma_{y}=2\sigma_{x}$, the weight is c=0.8 and the standard deviation of the estimator is $\sigma_{z}\approx 0.89\sigma_{x}$.

There are three main challenges in fusing IMU data. First, the sensor signals of the IMU, which are supposed to be fused, are correlated but do not have the same amplitude, which is denoted as *f* above. Therefore, the signals must first be transformed to equalize the amplitude. For this purpose, a linear transformation is proposed as it maintains the normal distribution of the noise. Second, the structural vibration measurement contains many oscillation modes simultaneously and the correlation as well as transformation between the signals depend on the mode. However, since each mode has a different frequency, it is possible to fuse signals by combining data with regard to the frequency. Third, the noise variance is required to calculate the optimal weights and needs to be estimated.

Figure 7 presents an overview of the sensor fusion algorithm. The first step in the sensor fusion algorithm is preprocessing, which involves discrete differentiation of the angular rate signals $\omega_{x}$, $\omega_{y}$ and $\omega_{z}$ to obtain the angular accelerations ${\dot{\omega }}_{x}$, ${\dot{\omega }}_{y}$ and ${\dot{\omega }}_{z}$. The signals are then decomposed using multi-level discrete wavelet transformation (MLDWT) [23], which is a key element and enables the frequency-dependent fusion of IMU data.

MLDWT is a wavelet-based analysis method for discrete signals, which extracts time-frequency features by decomposing time signals into low-frequency and high-frequency subseries on multiple levels. The MLDWT decomposes a discrete time signal into one approximation coefficient and *M* detail coefficients. Each coefficient represents time-frequency information of a unique range of frequencies. By fusing the wavelet coefficients of different signals at a specific level, signals can be fused with regard to frequency. The result of the fusion is a single wavelet coefficient. The appropriate mother wavelet for the MLDWT is determined experimentally by minimizing the error of a sample signal with vibrations and the transformed and recomposed version of the same signal. It is chosen to be a Coiflet wavelet with five vanishing points. Furthermore, the number of decomposition levels is $M=\lfloor \log_{2}\left(N\right)\rfloor$, where *N* is the length of the signal.

In the next step, data fusion is performed for each previously defined mode. The wavelet level $m_{u}$, which contains the frequency component $f_{u}$ of an oscillation mode *u*, is calculated as:(9)$$m_{u}=\left\lfloor \log_{2}\left(f_{s}/f_{u}\right)\right\rfloor ,$$
where *u* is the index of the mode, $f_{s}$ is the sampling rate and $f_{u}$ is the frequency of the oscillation mode. The level $m_{u}$ is then used to extract the relevant wavelet coefficients. Fusion at each oscillation mode requires some prior knowledge of the system, which is provided by the fusion array O(u). Each element of the fusion array contains a vector with six coefficients $t_{ax}$ to $t_{{\dot{\omega }}_{z}}$ and the frequency $f_{u}$ of the oscillation. The coefficients $t_{ax}$ to $t_{{\dot{\omega }}_{z}}$ describe the transformation between the signals. The coefficients are determined experimentally by measuring high amplitude vibrations and numerically estimating the transformation coefficients by minimizing the mean squared error (MSE).

A transformation coefficient of 1 is assigned to the main signal with the highest SNR. As a result, the other signals are transformed to match the amplitude of the main signal. Signals that are not correlated by the beam oscillation have a coefficient of zero and are removed by the Signal Selection Filter. An example for an element of the fusion array is shown in Equation (Equation 24). The transformation of wavelet coefficients is a multiplication with the transformation coefficient:(10)$$W\hat{S}\left(m_{u},n\right)=t_{S}Ws\left(m_{u},n\right),$$
where $WS(m_{u},n)$ is the wavelet coefficient of a signal from the IMU, $t_{S}$ is the corresponding transformation coefficient and $W\hat{S}\left(m_{u},n\right)$ is the transformed value. After transformation, uncorrelated signals are removed by the Signal Selection Filter, which removes signals with a total sum of zero. In the next step, signals are combined by the adaptive combiner. The optimal weights are calculated as:(11)$$c_{i}=\frac{\sigma_{i}^{-2}}{\sum_{i=1}^{I}\sigma_{i}^{-2}},$$
where *i* is the index representing the different signals for fusion, $c_{i}$ is the optimal weight for signal *i*, $\sigma_{i}$ is the standard deviation of noise of signal *i* and *I* is the total number of signals to fuse. The result is a single wavelet coefficient containing the combined sensor signals. The time signal of the fusion result is obtained using inverse MLDWT. The result is a time signal with a narrow frequency band, as it only represents the oscillation of a single oscillation mode.

The noise standard deviation $\sigma_{i}$ of each signal needs to be estimated. The implemented estimator is based on [24] and estimates the noise variance by calculating the median of the wavelet coefficients in a section where *f* is piecewise smooth. This is a section where little or no vibration occurs. The standard deviation of the noise is estimated to be:(12)$$\sigma_{i}=\frac{Med(|S_{i}|)}{0.6748},$$
where $Med(|S_{i}|)$ is the median of the signal $S_{i}$. The analysis of the sensor noise shows that the noise in the accelerometers has a uniform noise density and Gaussian distribution. In this case, the best estimate is calculated by analyzing the finest-scale wavelet coefficient. The noise in the gyroscope signals is unevenly distributed in the frequency domain and increases at higher frequencies. Because the noise is level-dependent, it must be estimated for each wavelet coefficient individually.

In the real-world scenario, piecewise smooth sections need to be identified to update the noise estimates. In the experimental measurements presented here, no piecewise smooth section is available for noise estimation. Therefore, the noise variance is estimated prior using a separate measurement without vibrations. It is provided as the noise matrix:(13)$$\sigma_{IMU}=\left[\begin{array}{ccc} \sigma_{a_{x}1} & \cdots & \sigma_{a_{x}M}\\ \vdots & \ddots & \vdots \\ \sigma_{{\dot{\omega }}_{z}1} & \cdots & \sigma_{{\dot{\omega }}_{z}M} \end{array}\right]$$

### 2.6. Experimental Setup

The proposed concept is validated experimentally by measuring structural vibrations of a husbandry cage using different sensor arrangements. The vibrations are generated with a custom-built force pulse generator. The experimental setup emulates activity-induced structural vibrations with force pulses that act vertically on the cage floor. The setup includes: mice cages of type Zoonlab HRC500 [22], 200 g of wood flocking, an aluminum stand for the cages, a force pulse generator, a heavy aluminum base to reduce external vibrations and several microcontrollers. The components are shown in Figure 8.

There is a wide selection of commercial multi-axial IMUs available such as the Bosch BMI270 [25], STMicroelectronics ASM330LHHTR [26] and the TDK InvenSense ICM20948 [27]. For this work, the ASM330LHHTR was chosen due to its superior noise characteristics and sampling rate. It has a noise density of 60 μg/$\sqrt{H}$z for the accelerometers and 0.005 dps/$\sqrt{H}$z for the gyroscopes, which is lower than the other IMUs mentioned. Furthermore, it has a maximum sampling rate of 6667 Hz for all sensors.

Two sensor arrangements are evaluated, which are shown in Figure 9. The first arrangement serves as reference and has an IMU directly mounted in the center of the cage. This location is deemed the optimal location in Section 2.3. The second cage has an IMU with the proposed beam structure. The beam uses the optimal design that is covered in Section 3.1. The prototype is milled from a piece of FR-4 PCB material. Both arrangements use the same STMicroelectronics ASM330LHHTR sensor.

Another parameter of the experiments is the location where the force pulses act on the cage. Two excitation locations are defined to investigate whether amplification also occurs when the cage is excited at different locations. The first location is the center of the cage floor. The second location is beside the center and about half the distance towards the short side of the cage. The locations are illustrated in Figure 8.

#### 2.6.1. Force Pulse Generator

The experiment requires a purpose build force pulse generator, which is able to generate very small force pulses. The force pulse generator uses a magnetic actuator and is driven with a rectangular signal of 12 V. The pulse amplitude is controlled by the duration of the rectangular control signal and is modulated between 0 s and 1000 μs. At 12 V, the actuator generates a constant force of 2.16 N, which amounts to a force pulse range of 0 to 2.16 mNs. The force pulse is calculated as:(14)$$|\vec{p}\left|=\right|\vec{F}|\Delta t,$$
where Δt is the duration of power supplied to the actuator and $|\vec{F}|$ is the force generated by the actuator. The repeatability error of the experimental setup is explored in Section 2.6.3.

For comparison, the pulse of a free-falling object can be calculated as $|\vec{p}\left|=m\right|\vec{v}|=m\sqrt{2gh}$ when air resistance is ignored. A 25 g mouse would therefore create a pulse of 7.83 mNs when it lands from a height of five millimeter.

#### 2.6.2. Experimental Measurements

The experimental evaluation aims to explore the characteristics of the proposed sensing device compared to a directly mounted sensor for different force pulse strengths and excitation locations. In the experiment, force pulse sweeps are used to investigate the aforementioned characteristics. The individual force pulses are spaced apart by 0.5 s, which allows each structural vibration to decay before the next starts. Each measurement begins with a starting sequence of four force pulses, after which 100 force pulses are generated at a rate of 2 Hz. The pulses go up in equidistant steps. In this work, several experiments are presented which have different amplitude ranges and location of force pulse application depending on the evaluation. A sample of a measurement is shown in Figure 10.

#### 2.6.3. Repeatability of Experimental Setup

The repeatability error of the experimental setup is assessed throughout the entire range of force pulse amplitudes. For this, twenty amplitude sweeps are recorded using an IMU without the beam structure. The amplitude sweep consists of 100 pulses from 21.6 μNs to 2160 μNs and the force pulses are located in the center. Based on the acceleration data, the vibration magnitude is calculated for each force pulse. Afterwards, the mean and standard deviation are calculated to determine the average magnitude and error. As a measure of magnitude, the energy of the acceleration magnitude $|\vec{a}|$ is calculated. The energy is calculated over a period of 0.2 s, which is the longest duration of a decaying vibration. The magnitude and energy are calculated as follows:(15)$$|\vec{a}|=\sqrt{a_{x}^{2}+a_{y}^{2}+a_{z}^{2}},$$
(16)$$E\left[i\right]=\sum_{n=1}^{N=\lfloor 0.2\;s\times f_{s}\rfloor }{\left|\vec{a}\left[n\right]\right|}^{2},$$
where E[i] is the energy at a certain force pulse *i*, *N* is the number of samples and $f_{s}$ is the sampling rate of the sensor. The mean and standard deviation of E[i] are shown in Figure 11.

The average vibration magnitude in the left plot of Figure 11 has a flat slope at the beginning that becomes steeper as it progresses. The error depicted in the plot in the middle follows a similar path and increases with the magnitude of the vibration. The relative error shown in the right plot is calculated as the fraction of error and average. It shows a declining trend with a variation of about ± 1%. In the beginning, the values average about 3% and decline to about 1% at the end.

The results illustrate that the experimental setup is capable of generating structural vibrations with a high degree of repeatability. Furthermore, it is observed that the energy of the vibration magnitude does not rise linearly with the force pulse amplitude. Although starting at about 1550 μNs, the average energy appears to enter a quasi-linear region.

## 3. Results

The performance of the proposed sensing device is evaluated based on measurements of structural vibrations, which are generated with a highly repeatable experimental setup. This allows for a direct comparison between a rigidly mounted sensor and the proposed IMU using a PCB beam structure. The results of the FEM simulation and experimental evaluations are presented in the following.

### 3.1. Beam Optimization

As described in Section 2.4, the structural vibration of the system of cage and beam structure is simulated to determine an optimal design of the beam that maximizes the vibration amplitude. The best result is achieved using a beam with a width of 3 mm and a length between 27 mm and 36 mm. The results are shown below and a simulation without a beam structure is included for reference. In the context of the beam structure, the width corresponds to the variable *w* and the length to *l* as described in Section 2.4.

Figure 12 illustrates the acceleration observed at the location of the virtual IMU. The *x*-axis depicts the different simulations. For each simulation, the bars depict the signal energy $E_{i}$ for the *x*-, *y*- and *z*-axis, respectively, as well as the total energy $E_{Total}$. The signal energies are calculated as:
(17)$$E_{i}=\sum_{n=1}^{N}a_{i}^{2},$$
(18)$$E_{Total}=E_{a_{x}}+E_{a_{y}}+E_{a_{z}},$$
where $a_{i}$ is one of the three acceleration signals and *N* is the number of samples. The bars show that most of the signal energy is contained in the *z*-axis. Starting from the left, the total signal energy rises up to the beam of length 30 mm, which has the highest signal energy. Compared to the simulation without a beam, the beam exhibits a 1.72-times-higher signal energy.

From these results, it is concluded that the beam structure with a width of 3 mm and a length of 30 mm exhibits the highest signal energy, which makes it the most optimal design to achieve a high SNR.

### 3.2. Vibration Spectrum in Simulation

The four beams, from Section 3.1, with the highest signal energy are analyzed further by Fourier transforming the acceleration data. The spectrum of acceleration for the the *x*-, *y*- and *z*-axis, respectively, is shown for beams with a length 29 mm to 32 mm in Figure 13.

The spectrum of the IMU without a beam only exhibits a few small peaks in the z-axis. The results using a beam structure show multiple peaks in all axes. Between these results, the peaks exhibit similar frequency and magnitude. It is observed that a longer beam results in oscillation of lower frequency. Due to the resolution of the Fourier transform, the exact amplitude at the peak can only be estimated, but the beams with length 29 mm and 30 mm exhibit the highest amplitude at a frequency of around 102 Hz.

The spectra of simulations with a beam depict the deformation forms described in Section 2.4. The big peak in the *z*-axis and the smaller peak in the *y*-axis around 100 Hz correspond to the first deformation form. The peak at 400 Hz in the *x*-axis corresponds to the second deformation form. Since the sensor is in the axis of rotation of the third form, there is no acceleration peak at 700 Hz. Frequency components corresponding to the fourth form are not visible in the spectrum.

Comparing the 30 mm beam to the simulation without a beam, based on the amplitude of the highest peaks, a 2.4-times-higher amplitude is observed with the beam. All in all, the simulation shows that the signal amplitude and energy are amplified by using a beam to sense vibration.

### 3.3. Experimental Vibration Amplification

The experimental evaluation involves multiple measurements with force pulse sweeps from 21.6 μNs to 2160 μNs, where two characteristics are explored. The first is the amplification of vibration achieved by the proposed sensing device. For this, measurements using an IMU without a beam are compared with measurements of an IMU with the beam structure. Secondly, the location-dependency of the vibration amplification is explored by changing the location of the force pulses between the center and an off-center location shown in Figure 8. Twenty measurements are recorded for each combination of sensor arrangement and excitation location for a total of 80 measurements.

In the first assessment, the vibration magnitude is evaluated. It is calculated as the energy of the acceleration magnitude. The acceleration magnitude and energy are calculated as:(19)$$|\vec{a}|=\sqrt{a_{x}^{2}+a_{y}^{2}+a_{z}^{2}}.$$
(20)$$E(|\vec{a}\left|\right)=\sum_{n=1}^{N=\lfloor 0.5\;s\times f_{S}\rfloor }{\left|\vec{a}\right|}^{2}\left[n\right].$$
where $f_{S}$ is the sampling rate. The magnitude of very small vibrations is significantly influenced by the sensor noise. In Section 2.5, the sensor noise is modeled as additive Gaussian noise with zero mean. Using the characteristic function of a Gauss distribution, it is found that the acceleration magnitude $|\vec{a}|$ also has additive Gaussian noise. Therefore, the true signal energy is estimated according to the following equations:(21)$$X=|\vec{a}|+\sigma ,$$
(22)$$E\left(X\right)=\sum X^{2}=\sum |\vec{a}{|}^{2}+\sum \sigma^{2}+\sum 2\left|\vec{a}\right|\sigma ,$$
(23)$$E(|\vec{a}|)\approx E\left(X\left[n\right]\right)-E\left(\sigma \left[n\right]\right),$$
where *X* represents the measurement, $|\vec{a}|$ represents the true acceleration and σ represents the additive sensor noise. Equation (Equation 22) shows that the energy of the measurement is comprised of the energy of the desired value $E(|\vec{a}|)$, the energy of the noise E(σ[n]) and a remainder $\sum 2|\vec{a}|\sigma$. Since σ has zero mean, the remainder is small compared to E(X[n]) and E(N[n]). Therefore, it is neglected and the true vibration magnitude is estimated by subtracting the energy of the noise. In the experiment, the energy of the noise is estimated in a section, where no vibrations occur.

The results in Figure 14 show the average noise-compensated signal energy of the acceleration magnitude. The results for an IMU without a beam and an IMU with a beam are shown in different colors and the *x*-axis depicts the force pulse amplitude. The graph on the left shows the results with force pulses located in the center and the graph on the right shows results for force pulses off-center. The error bars depict the standard deviation. The amplification, which is the ratio of the measurements with a beam and without a beam, is shown as a percentage with black diamonds. The percentages are shown on the right vertical axis.

Both results in Figure 14 depict an amplification of vibration when using the proposed beam structure. In plot (1), where force pulses are applied in the center, a maximum amplification of 221% is observed at a force pulse of 713 μNs. In plot (2), where force pulses are applied off-center, a maximum amplification of 480% is observed at a force pulse of 1512 μNs.

In order to evaluate the oscillation modes in the measurements, the Fourier transformed acceleration and angular rate are evaluated in the following. In Figure 15, the frequency spectra of measurements with and without the beam are shown, where force pulses are applied in the center. For the IMU without a beam, vibration is only observed in the z-axis and it is distributed over a large frequency range. For the IMU with a beam, many distinct peaks are observed in all but the z-axis of the gyroscope. Comparing the largest peaks in the acceleration z-axis, the measurement with a beam structure shows a 4.4-times-higher amplitude.

Based on the oscillation forms shown in Section 2.4 and the peaks in the spectrum depicting the measurements with a beam, it is hypothesized that certain signal components are correlated. These are the signal components in $a_{z}$ and $\omega_{x}$ at 100 Hz and $a_{y}$ and $\omega_{x}$ at 1000 Hz.

Figure 16 illustrates the results of measurements with force pulses off-center. Comparing the largest peaks in the acceleration *z*-axis, the measurement with a beam structure shows a 6.2-times-higher peak amplitude over the measurement without a beam.

The spectra in Figure 15 and Figure 16 show some differences. In the measurements with a beam structure, the peaks are at a similar amplitude, but the peaks at 1000 Hz and 450 Hz are less pronounced. In the measurements without a beam, the overall amplitude is smaller and higher frequency components are reduced significantly.

### 3.4. Sensor Fusion in Experiment

The sensor fusion algorithm combines the six sensor signals $a_{x}$, $a_{y}$, $a_{z}$, $\omega_{x}$, $\omega_{y}$ and $\omega_{z}$ of the IMU to reduce the intrinsic noise. For fusion, the angular rate is differentiated to obtain the angular acceleration ${\dot{\omega }}_{x}$, ${\dot{\omega }}_{y}$ and ${\dot{\omega }}_{z}$.

Combining the findings of Section 2.4 and the spectra in Section 3.3, two oscillation modes are identified for sensor fusion. The first oscillation mode is at 103 Hz and creates a correlated oscillation of $a_{z}$ and ${\dot{\omega }}_{x}$, where *a* refers to acceleration and $\dot{\omega }$ refers to angular acceleration. Examining further, the oscillation at 103 Hz actually consists of two modes at 90 Hz and 103 Hz. These two modes have the same oscillation form and exhibit very similar transformation coefficients. In addition, they fall into the same wavelet level due to their close proximity frequency-wise. For the purpose of sensor fusion, they are treated as one oscillation mode. The second oscillation mode is at 1000 Hz and involves oscillation of $a_{y}$ and ${\dot{\omega }}_{x}$. The information describing the oscillation modes is stored in the fusion array:(24)$$O\left(1\right)=\{\left[0,0,1,-7.2\times 10^{-5},0,0\right],103\mathrm{Hz}\},$$
(25)$$O\left(2\right)=\{\left[0,1,0,-6.8\times 10^{-6},0,0\right],1000\mathrm{Hz}\},$$
where *O* depicts the fusion array. In the first element O(1), the main sensor signal is $a_{z}$ and ${\dot{\omega }}_{x}$ is correlated by a factor of $-7.2\times 10^{-5}$. In O(2), the main sensor signal is $a_{y}$ and ${\dot{\omega }}_{x}$ is correlated by a factor of $-6.8\times 10^{-6}$.

The sensor fusion performance is evaluated experimentally by using the algorithm on a test measurement and examining the noise characteristics before and after the sensor fusion has been used. The test measurement has a force pulse amplitude sweep from 6.48 μNs to 648 μNs. The fusion algorithm requires noise estimations, which are obtained as described in Section 2.5 using another measurement where no vibrations occur. Equations (Equation 7) and (Equation 8) are used to calculate the expected noise reduction. The noise estimates are given in Table 1.

In order to estimate the noise contained in the experimental measurement, a wavelet-based denoising method is used to separate the vibration and noise signal. The standard deviation of the noise signal is then calculated to evaluate the noise level. The denoising method uses a threshold filter in the wavelet domain and is based on [24]. The method applies a hard threshold to the wavelet coefficients of a signal to remove the noise. The threshold value *T* is based on the Universal Threshold proposed in [28] and is calculated as follows:(26)$$T=\sigma \sqrt{2\log N}\times 0.67,$$
where *T* is the threshold value, σ is the theoretically calculated standard deviation of the noise and *N* is the length of the wavelet coefficient. The factor of 0.67 is determined manually to extract most of the vibration components. The noise signal is obtained by subtracting the denoised signal from the original signal. The different signals that are calculated are shown in Figure 17.

A sample of 2 s is shown in Figure 17. The rows of the plots show the result of the two oscillation modes, while the columns illustrate the unfiltered signal, the wavelet-denoised signal and the noise signal. In the first row of plots, the signals with sensor fusion appear less noisy than the original signal, as observed visually. In the second row, the signals with and without sensor fusion show little visual difference at this scale.

Based on the noise signals, shown in the third column, the noise is estimated by calculating the standard deviation. For the oscillation at 103 Hz, the signal without sensor fusion has a noise of 0.66149 × 10^−3^ g and 0.6246 × 10^−3^ g with fusion. This corresponds to a noise reduction of 0.0369 × 10^−3^ g or 5.573%. For the second oscillation at 1000 Hz, the noise standard deviation is 18.782 × 10^−3^ g without fusion and 18.738 × 10^−3^ g with fusion. This corresponds to a noise reduction of 0.044 × 10^−3^ g or 0.23%. In comparison to the values in Table 1, the noise reduction achieved in experiment is smaller than theoretically calculated.

### 3.5. Influence of Ambient Temperature

A change in ambient temperature affects the mechanical properties of the beam structure and characteristics of the sensing device. However, temperature fluctuations are very unlikely in this use case scenario because the room temperature is kept constant for the wellbeing of the animals. The temperature influence is investigated in simulation by evaluating the signal energy at the location of the virtual IMU for different temperatures.

Figure 18 illustrates the vibration characteristics of a sensing device with a beam of l=30mm and w=3mm. The vibration energy is practically unchanged in a range of −30 °C to 20 °C. At higher temperatures, the signal energy decreases.

## 4. Discussion

The results illustrate that the proposed concept achieves the desired effect of amplifying structural vibration to increase the SNR. The results also illustrate a number of other features, which will be discussed in more detail here.

### 4.1. Experimental Setup

In Section 2.6.3, it is shown that the experimental setup is capable of generating structural vibrations with a high degree of repeatability. Based on the energy of the vibration, an error of about 1% up to 3% is observed depending on the amplitude of the force pulse. The experimental setup is sufficient to evaluate the vibration amplification of the proposed concept, where differences of several hundred percent are observed.

While the repeatability error is small, the actual amplitude of the force pulses is not calibrated. A calibrated force pulse generator is useful to assess the correlation between force pulse strength and structural vibration magnitude, which is not subject of this work. Because of the short duration of the force pulses, a force measurement method with a high sampling rate is needed. Piezo-electric force transducers have a sampling rate in the upper kilohertz range.

### 4.2. Optimal Beam Design

The experiment shows a greater amplification than the simulation results. In the spectrum, the simulation predicts an amplification of 2.4 and the experiment shows an amplification of 4.4 times. In terms of signal energy, an amplification of 221% is observed in experiment and the simulation predicts an amplification of 172%. The cause of these differences are inaccuracy of parameters and idealizations in the simulation model, which are discussed in detail in Section 4.4.

The results indicate that the simulation setup is sufficiently accurate to design a beam structure with the intended vibration amplification characteristics. However, when employed in the real world, these designs may not perform optimally. A proposed solution to guarantee optimal vibration amplification is the implementation of a tuning component. It is used to adjust the oscillation characteristics in the operating environment in order to adapt to the oscillation characteristics of the real-world environment. A tunable weight is proposed as a tuning component, which is mounted at the end of the beam. The beam characteristics are adjusted by changing the mass of the weight or the position of the weight. Another approach is to mount the beam using laterally oriented slits instead of round holes. This allows for adjustment of the free-standing beam length and in turn adjust the effective length of the beam. A longer beam results in a lower oscillation frequency. However, this type of adjustment requires recalibration up on reassembly and is very tedious in practice.

### 4.3. Vibration Amplification in Application

In the experimental evaluation, a significant amplification of vibration is observed. The evaluation in Section 3.3 shows that the desired effect is achieved whether the cage is excited in the center or closer to the walls of the cage. This is important in the application, as it increases the sensitivity of the system in all scenarios. Furthermore, it is observed that the amplification is higher when the cage is excited off-center. This effect is very beneficial, as force pulses at the periphery generally generate weaker vibrations. In the application, this could help in the detection of physical activity in the periphery. Furthermore, when comparing Figure 15 and Figure 16, the composition of oscillation modes changes depending on the location of the force pulse. In the future, this phenomenon could be used to extract location-specific information by analyzing the composition of the oscillation modes.

The correlation between amplification and vibration magnitude is investigated in Section 3.3 and the results are shown in Figure 14. In this examination, a series of force pulses generate vibrations with increasing magnitude, starting with vibrations with a magnitude below the sensor noise. It is evident that the sensing device produces an amplification effect throughout the tested amplitude range. It is observed that the amplification is highest at lower amplitudes, which supports the intended use of amplifying very small vibrations that would otherwise be masked by noise. At higher magnitudes, the amplification is smaller but still very significant at over 165%. At smaller amplitudes, the deformation of a bending beam is approximately linear. At higher amplitudes, the deformation causes non-linear effects, which are believed to reduce the amplification.

### 4.4. Deviation of Simulation and Experiment

In Section 3.3, the experimental data of a beam using the optimal design from Section 3.1 is obtained. The experimental data are compared to the simulation results in the frequency domain, where signal peaks are a representation of different oscillation modes. The comparison shows many similarities such as the peaks in $a_{z}$ at around 100 Hz, the peaks in $a_{y}$ at around 100 Hz and the peaks in $a_{x}$ at around 400 Hz. However, the exact frequency and amplitude of the peaks differs between experiment and simulation. Furthermore, the experimental data show additional signal components in the $a_{y}$ axis at 900 Hz to 1000 Hz, which are not present in the simulation results. Based on the modal analysis in Section 2.4, these correspond to the oscillation type titled as fourth oscillation form.

The differences between simulation and experiment are attributed to three main causes. First, the material parameters like the Young’s modulus and damping coefficient of the cage and the beam are estimated using standard values from a library. The inaccuracy of these parameters is accepted, because more accurate estimations would require extensive and destructive testing, which was not feasible. Another inaccuracy is the weight of the sensing board, which fluctuates due to the inconsistent amount of solder used for mounting. Secondly, the simulation model is idealized in some aspects to reduce computational cost and complexity. The wood flocking used to fill the floor is simulated as a surface mass, which neglects damping properties. The data lines that are milled into the beam are able to reduce the mechanical stiffness but were neglected. The third cause of disparity are the tolerances in manufacturing and assembly. The milling operation has good tolerances, which have minimal influence on the oscillation characteristics of the beam. The assembly is conducted by hand, which has an error of about 1 mm in the worst case. However, the assembly error alone does not explain the observed deviation, since beams with lengths between 29 mm and 32 mm show less deviation between each other in the simulation than compared to the prototype beam.

### 4.5. Noise Reduction with Sensor Fusion

In Section 3.4, the proposed sensor fusion algorithm is successfully validated by fusing accelerometer and gyroscope data in an experiment. The IMU data are first preprocessed to obtain the acceleration and angular acceleration for each spatial axis, which are then selectively fused to produce a single signal with reduced noise.

The sensor fusion algorithm achieves a noise reduction of 5.573% for the oscillation at 103 Hz and 0.23% at 1000 Hz in the experiment. The theoretical calculation predicts a noise reduction of 6.9% at 103 Hz and 0.276% at 1000 Hz in an ideal case, which is higher than the results of the experiment. This is explained by the error of estimated parameters, which reduces the effective noise reduction. Parameters that are estimated are the noise levels and the transformation coefficients. The experiment shows that the proposed sensor fusion algorithm achieves the desired effect. Furthermore, the noise reduction in the experiment approaches the theoretical maximum, which shows that the estimated parameters are sufficiently accurate.

Table 1 shows that the noise of the angular acceleration is significantly higher than the acceleration. As explained in Section 2.5, the maximum noise reduction is achieved when the noise level is equal in all signals. To maximize the noise reduction, the amount of angular rotation during oscillation must be increased for a higher snr in the angular acceleration sensor signals. The relationship between the translation *z* and rotation θ for a point load at the end of the beam is:(27)$$\theta =\frac{3}{2l}z,$$
according to [21]. For a uniform surface load, the relationship is:(28)$$\theta =\frac{4}{3l}z.$$

These equations indicate two methods of increasing the angular acceleration. First, the rotation is anti-proportional to the length of the beam. Therefore, a shorter beam should create more rotation at a given translation *z*. Secondly, the factor that is dependent on the type of load is more advantageous for a beam with a point load at the end. This means that most of the mass of the free-standing beam should be concentrated at the end. Further, it may even be advantages to place a mass at the end of the beam to move the center of mass further towards the end.

### 4.6. Limitations

In the discussion of the proposed concept, two main limitations stand out. First, the beam structure needs to be redesigned and adapted to each new application because the beam has to be tuned to the specific vibration of the ambient object. A possible solution is given in Section 4.2. Using a tuning component, the vibration characteristics of the beam can be adjusted in the application environment to achieve vibration amplification. Secondly, the proposed beam structure massively alters the vibration characteristics and is probably unsuited for unbiased quantitative vibration analysis, e.g., seismic exploration. For vibration-based activity estimation, however, physical quantity is not as important because the relationship between activity and vibration is derived by analyzing exemplary datasets. The proposed sensing device could also be beneficial in applications where detection of vibration is the main goal.

### 4.7. Comparison of SNR

The signal-to-noise ratio at a given force pulse amplitude can be derived from the results in Figure 14. The SNR is equal to the ratio of the signal energy of the measurement and the energy of the noise, which is about 0.1 g${}^{2}$. At a force pulse of 1512 μNs in the experiment where force pulses are applied off-center, the IMU without a beam shows a signal energy of 0.51 g${}^{2}$ and the IMU with the beam structure an energy of 2.46 g${}^{2}$. These values correspond to an SNR of 5.1 and 24.6, which clearly illustrates the benefit of the proposed beam structure.

A direct comparison to geophones needs to take the frequency of the signal into consideration, due to the difference in the frequency response of the sensors. In order to compare the IMU results to a geophone, SNR is calculated as the ratio of signal amplitude of the acceleration in the *z*-axis at 100 Hz as shown in Figure 16. The IMU without a beam shows an amplitude of $8.7\times 10^{-4}$ m/s${}^{2}$ and a noise of $1.86\times 10^{-4}$ m/s${}^{2}$, which amounts to an SNR of 4.7. The IMU with a beam shows an amplitude of $3.9\times 10^{-3}$ m/s${}^{2}$ and a noise of $1.85\times 10^{-4}$ m/s${}^{2}$, which amounts to an SNR of 21.1.

The noise floor of a geophone is estimated by the voltage noise of the amplifying electronics, which is the main source of noise [29]. The noise of the amplifier varies between models but is about 0.3 μV on average [30]. The commonly used SM-24 from sparkfun has a sensitivity of 28.8 V/m/s, which amounts to a noise of $1\times 10^{-8}$ m/s. At a frequency of 100 Hz, this is equivalent to an acceleration noise of $6.6\times 10^{-6}$ m/s${}^{2}$. Assuming that the geophone would show a similar amplitude as the IMU without a beam, the SNR is estimated to be 131.4.

The comparison shows that the proposed sensing device massively increases the SNR compared to a directly mounted IMU. While the geophone has a 28-times-higher SNR over a directly mounted IMU, the geophone SNR is only 6.2 times higher than the proposed sensing device, which makes it much more competitive to geophones. Furthermore, this comparison ignores the fact that high-frequency signals above 250 Hz can only be measured by an IMU.

## 5. Conclusions

In this work, a novel method is proposed to enhance vibration sensing with IMUs using an optimized beam structure. It presents the design process for the beam structure, as well as a sensor fusion algorithm for combining the multiple sensor signals of an IMU to reduce the intrinsic noise. The concept is evaluated in an experimental use case where vibrations of a husbandry cage are measured. The results show that a significant increase in signal amplitude and signal energy is achieved by the proposed beam structure. In addition, the amplification is independent of the location of the excitation and the magnitude of the vibration. Amplification is therefore achieved in all activity related scenarios of the application. In terms of amplitude in the frequency domain, up to 6.2 times amplitude amplification is achieved. In terms of signal energy, a maximum amplification of 480% is observed. Furthermore, the experimental evaluation of the sensor fusion algorithm shows a noise reduction of 5.6% when fusing the acceleration in the z-axis and angular acceleration around the *x*-axis at 103 Hz.

In summary, this work presents a new sensor device using a six-axis IMU that achieves a significant amplification of structural vibrations, increasing the SNR and enabling the detection of smaller vibrations. It is experimentally validated that the desired effect is independent of the location of excitation and magnitude of vibration.

## 6. Future Work

In future work, the performance of the beam structure will be further optimized by integrating a tuning component to adjust the vibration characteristics in the operating environment. This allows for optimal performance in the real world by compensating for inaccuracies. A promising approach is an adjustable weight at the end of the beam, which can be changed or moved along the beam.

The sensor fusion algorithm shows promising results in the experiment. Two approaches are proposed to increase the effectiveness of the sensor fusion algorithm in the future. First, the beam is optimized to produce more rotation at a given deflection, which increases the SNR of the gyroscope measurements. Secondly, the number of oscillation modes of the beam suitable for sensor fusion is increased.

Vibration-based activity monitoring applications are differentiated by the observed subject, the environment and the information derived from the structural vibration. In the future, our proposed sensing device will be transferred to other applications, where it can provide a much higher sampling rate and multi-dimensional vibration data at a lower cost when compared to conventionally used geophones.

## Figures and Tables

**Figure 1 sensors-23-08045-f001:**
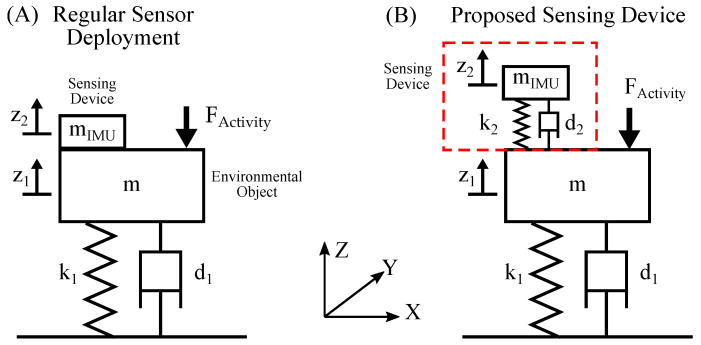
This figure illustrates the generalized physical model of the proposed sensing device. Subfigure (**A**) shows a regular directly mounted IMU and (**B**) shows an IMU with an added spring mass damper system as proposed.

**Figure 2 sensors-23-08045-f002:**
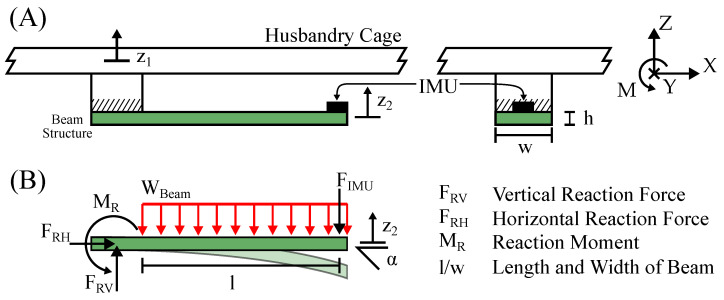
This figure illustrates the forces acting on the beam structure and IMU of the sensing device. In (**A**), a conceptual drawing shows a sensing device mounted on the floor of a husbandry cage. In (**B**), a mechanical drawing illustrates the forces acting on the beam.

**Figure 3 sensors-23-08045-f003:**
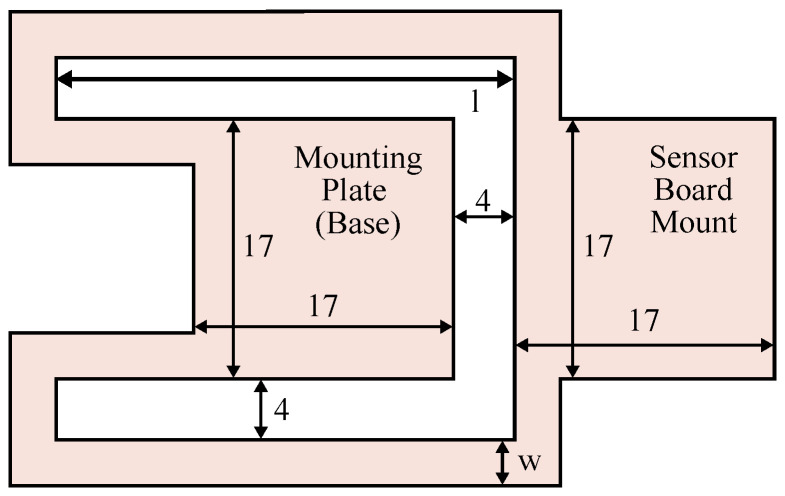
Technical drawing of the beam structure design.

**Figure 4 sensors-23-08045-f004:**
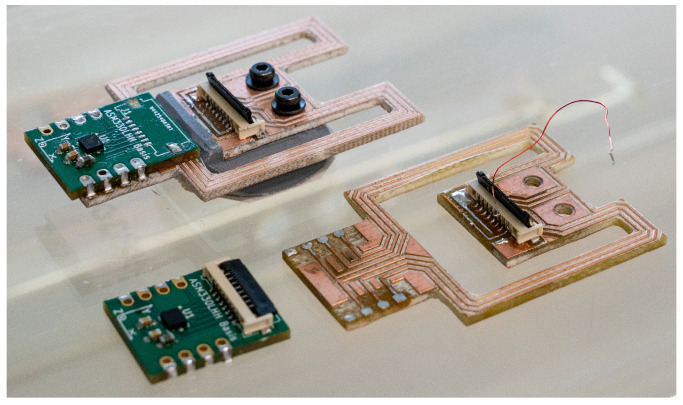
This figure shows two different beam structures and two sensor boards with the ASM330LHHTR 6-axis IMU. The upper beam is fully assembled and has braces with a width of 3 mm and a length of 30 mm. The lower beam has braces with a width of 3 mm and a length of 27 mm.

**Figure 5 sensors-23-08045-f005:**
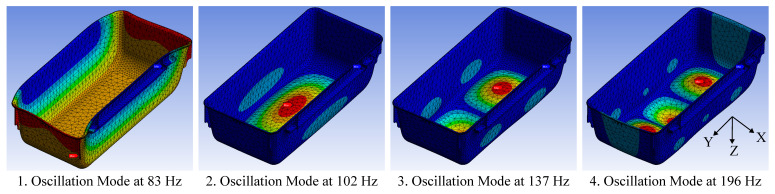
This figure illustrates the first four oscillations modes of the husbandry cage from a modal analysis.

**Figure 6 sensors-23-08045-f006:**
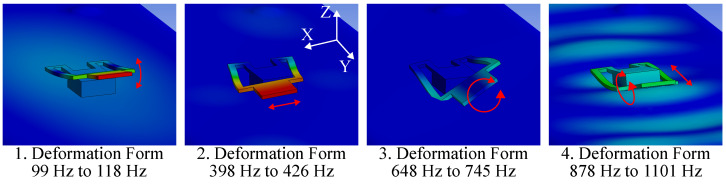
This figure shows four types of oscillation of a sample beam structure. The beam has braces with a width of 3 mm and a length of 29 mm. The deformation is shown exaggerated and the red arrows visualize the displacement.

**Figure 7 sensors-23-08045-f007:**
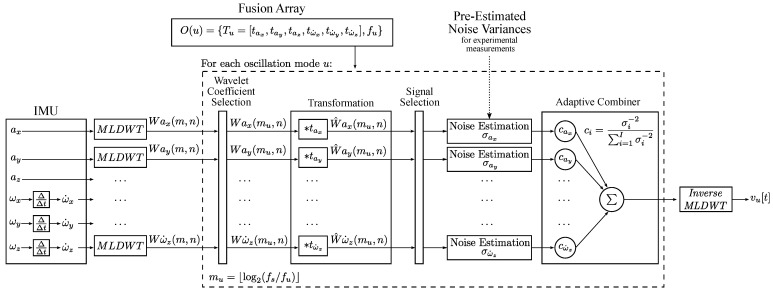
This diagram illustrates the structure of the sensor fusion algorithm.

**Figure 8 sensors-23-08045-f008:**
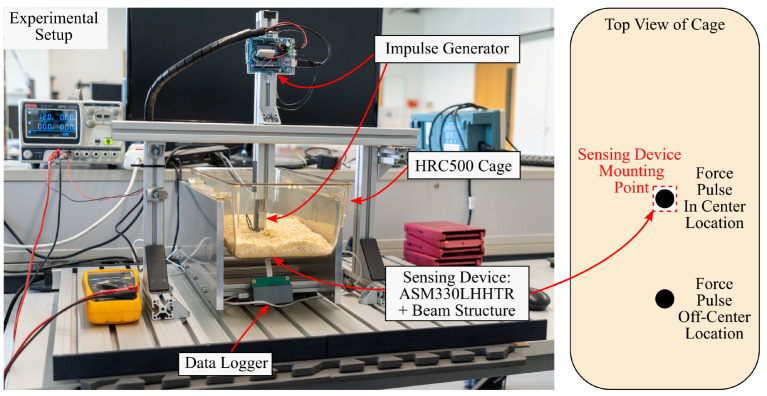
This figure shows the experimental setup. It depicts the force pulse generator, the cage stand, the base plate and a mice cage. On the right, the location of the sensor and the force pulses are shown as a schematic.

**Figure 9 sensors-23-08045-f009:**
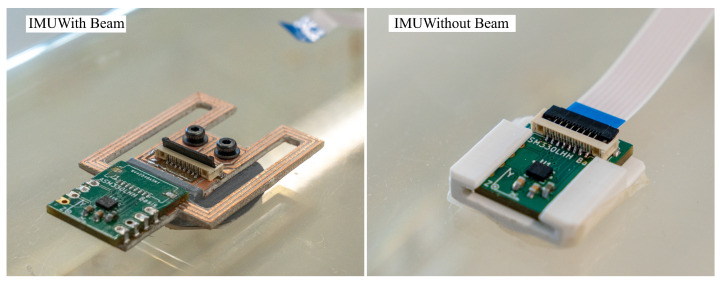
This figure shows the two sensor arrangements that are evaluated in the experiment.

**Figure 10 sensors-23-08045-f010:**
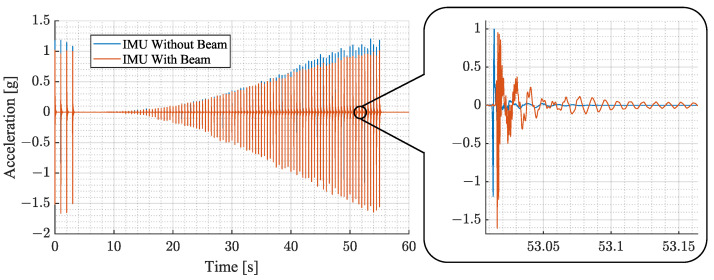
This figure shows a sample of the experimental measurements. It depicts the magnitude of acceleration over a span of 60 s. The plot in blue represents results of an IMU directly mounted to the cage without a beam. The plot in orange depicts the results using an IMU with the proposed beam structure.

**Figure 11 sensors-23-08045-f011:**
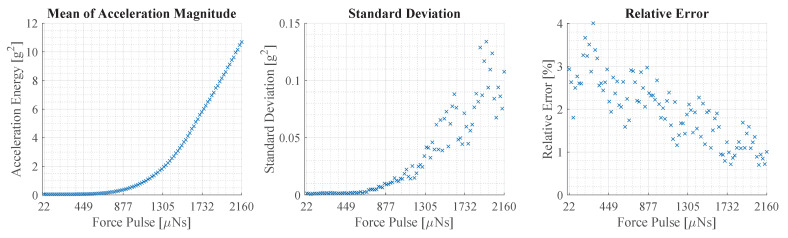
This figure illustrates the repeatability error of the experimental setup. The left plot displays the average vibration magnitude of 20 measurements at force pulses between 21.6
μNs and 2160 μNs. The plot in the middle shows the standard deviation. The plot on the right shows the relative error, which is the standard deviation divided by the average.

**Figure 12 sensors-23-08045-f012:**
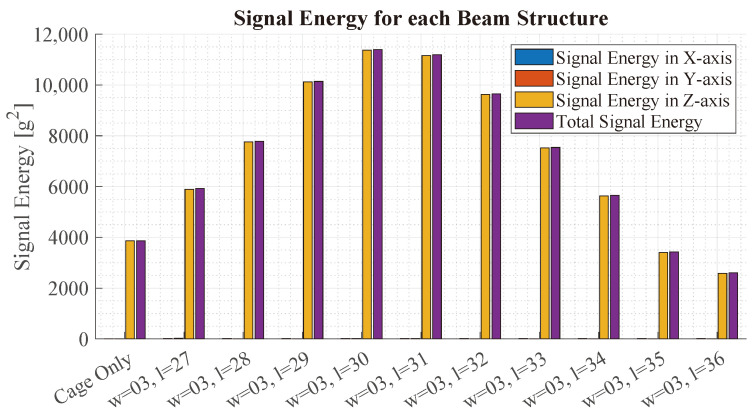
This figure shows the signal energy of the IMU using beam structures with different lengths from 27 mm to 36 mm in the FEM analysis. The bars illustrate the signal energy in the *x*-, *y*- and *z*-axis, respectively, as well as the energy of the magnitude. All beams have a width of 3 mm.

**Figure 13 sensors-23-08045-f013:**
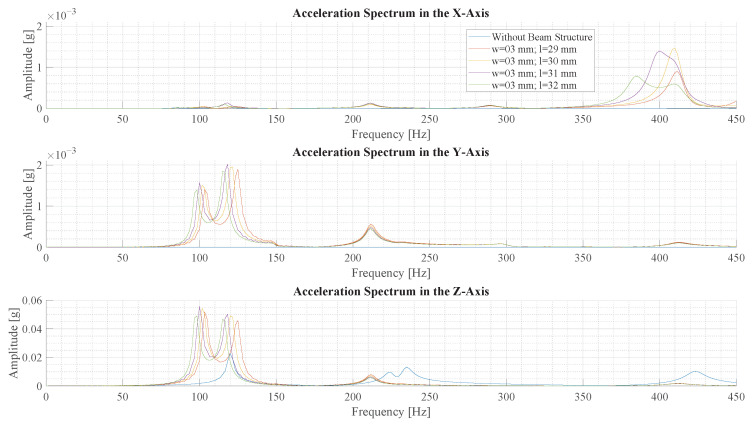
This figure illustrates the frequency spectrum of vibrations with different beam structures in a transient structural simulation. The vibration is created by a simulated force pulse of 150μN s.

**Figure 14 sensors-23-08045-f014:**
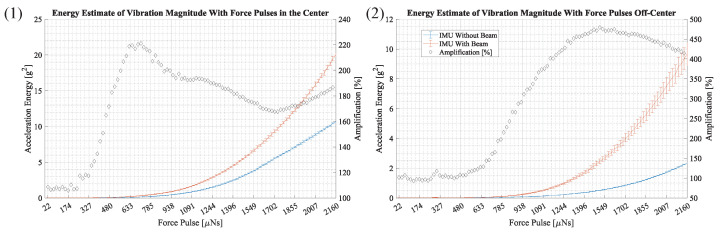
This figure shows the average energy of the acceleration magnitude at force pulses from 21.6 μNs to 2160 μNs. It depicts measurements of an IMU without a beam in blue and an IMU with a beam structure in orange. The error bars depict the standard deviation. The black diamonds illustrate the amplification in percent, which is the ratio of signal energy from the IMU with a beam and the IMU without a beam. The percentages are shown on the right vertical axis. Figure (**1**) and (**2**) depict results for force pulses in the center and off-center.

**Figure 15 sensors-23-08045-f015:**
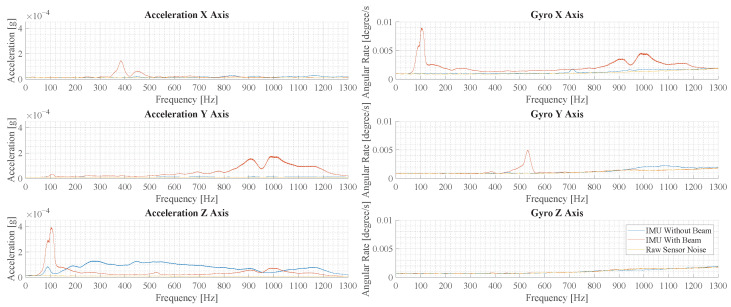
This figure illustrates the results of the experimental vibration measurement for force pulses in the center of the cage floor. Depicted are six plots that show the spectrum of the accelerometers and gyroscopes in the three spatial directions.

**Figure 16 sensors-23-08045-f016:**
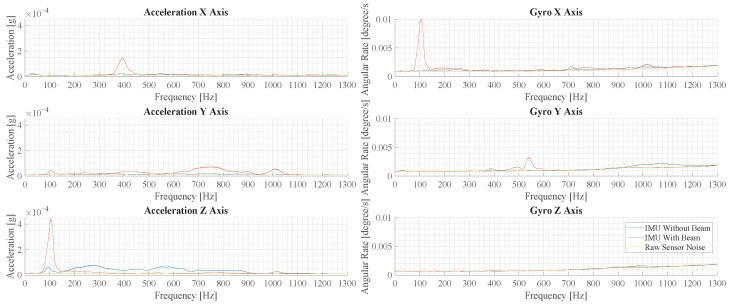
This figure illustrates the results of the experimental vibration measurement for force pulses off the center of the cage floor. Depicted are six plots that show the spectrum of the accelerometers and gyroscopes in the three spatial directions.

**Figure 17 sensors-23-08045-f017:**
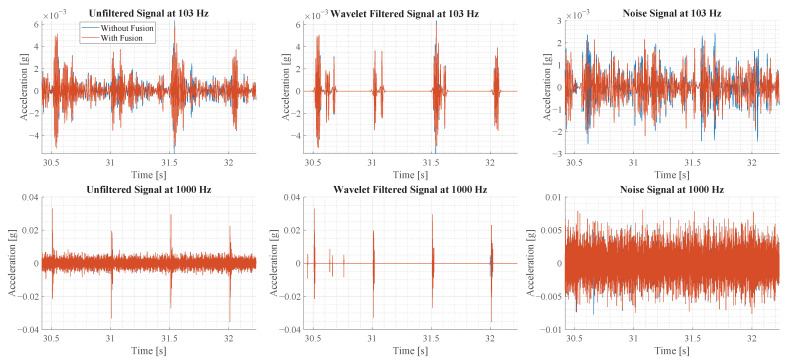
This figure illustrates the sensor signal before and after the sensor fusion for two oscillation modes. Furthermore, for each oscillation mode the original signal, the wavelet denoised signal and the noise signal are illustrated.

**Figure 18 sensors-23-08045-f018:**
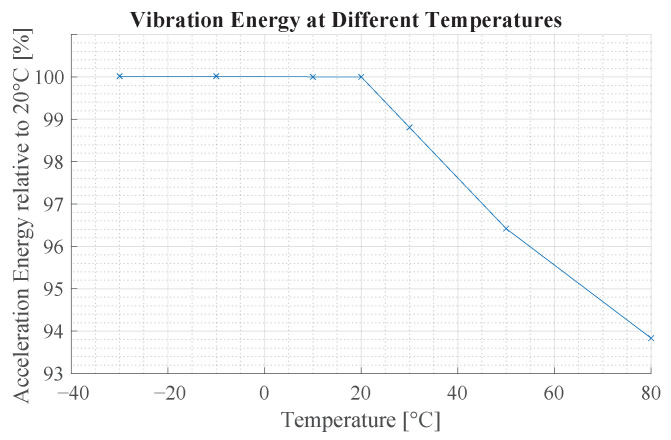
This figure shows the influence of temperature on the vibration characteristics of the beam structure. The y-axis shows the relative vibration energy compared to 20 °C of a beam with l=30mm and w=3mm.

**Table 1 sensors-23-08045-t001:** Noise estimates derived from an experimental measurement and the predicted noise reduction.

	O(1) at 103 Hz	O(2) at 1000 Hz
**Noise of Main Signal**	5.306 × 10^−3^ g	3.771 × 10^−3^ g
**Noise of 2nd Signal**	13.532 × 10^−3^ g	50.534 × 10^−3^ g
**Estimated Noise of Fusion**	4.9398 × 10^−3^ g	3.761 × 10^−3^g
**Noise Reduction**	0.37 × 10^−3^ g or 6.9%	0.01 × 10^−3^ g or 0.276%

## Data Availability

The majority of data processing is performed in matlab. The matlab scripts for data analysis and sensor fusion can be found here: https://github.com/pietertry/VibrationAmpBeamStructure, accessed on 8 September 2023.

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
