# Peer review of "A Vibration Sensing Device Using a Six-Axis IMU and an Optimized Beam Structure for Activity Monitoring"

_sensors, 2023, doi:10.3390/s23198045_

Round 1

Reviewer 1 Report

Dear Authors!

I suggest the following to help improve the quality of the article, please:

Define abbreviations in the abstract.

Enlist key contributions of the work in bullet form at the end of introduction section.

Is there a need to cite the reference for figure 1, please recheck.

is there a need to give reference to Figure 4, Please recheck.

Figure 7 seems ok but do Include formal algorithm in line with this figure.

In Figure 8, try to label each component/part of the hardware. This effort will make potential authors to understad this implementation very easily.

Is there a need to cite a reference for Figure 9, please recheck.

In line 528, instead of writing "following figure" its recommended to mention the figure number.

At lines 666 and 667, the word "achieves" has been repeated. Please proofread the whole article .

Include limitations of the proposed work.

Remove line 687 and 688.

Please provide data availability statement.

Remove dots from line 706.

Reviewer 2 Report

This paper proposes  A Vibration Sensing Device Using a 6-axis IMU and an Optimized Beam Structure for Activity Monitoring. In general, this paper is well presented. The following issues can be further considered.

1. More background and motivation of this study can be added, in case the readers are not very familiar with the topic.

2. The descriptions of the well known knowledge can be properly reduced.

3. Why introducing the IMU based method for the problem? What is the major benefits compared with traditional methods?

4. Some related works on this topic should be reviewed, such as "Data privacy preserving federated transfer learning in machinery fault diagnostics using prior distributions", "Intelligent Machinery Fault Diagnosis With Event-Based Camera", etc.

5. A couple of ablation studies should be added to evaluate the effects of the key parameters of the proposed method on the performance.

Reviewer 3 Report

This is a very interesting paper. There are several points that need improvements

1) is the idea really new? Mounting the IMU on a flexible beam seems like a novel idea at first, but considering that in real applications, IMU are always mounted on elastomers or some king of flexible support to prevent vibrations from saturating ADC, or because the environment is cluttered, I think that the situation is not new. The authors should comment on this.

2) the equations that could define the transfer function are given, but I was disappointed to see that nothing is done from them. I would expect to read something about the role of optimal placement of resonance mode in the Bode plot.

3) the paper is too long. This is detrimental to the convey of the ideas. The application part is important but needs shortening (a lot).

I would suggest to related this work to similar situation where flexibilities impact IMU readings (see below and references therein). 

M. Vigne, A. El Khoury, M. Pétriaux, F. Di Meglio and N. Petit, "MOVIE: a Velocity-aided IMU Attitude Estimator for Observing and Controlling Multiple Deformations on Legged Robots", in Proc. of the 2022 IEEE International Conference on Robotics and Automation

M. Vigne, A. El Khoury, F. Di Meglio, and N. Petit, "State Estimation for a Legged Robot with Multiple Flexibilities using IMUs: a kinematic approach", in IEEE Robotics and Automation Letters (RA-L), Vol. 5, Issue 1, pp. 195--202, 2020

no issue.

Reviewer 4 Report

The paper works on  Vibration Sensing Device Using a 6-axis IMU and an Optimized Beam Structure for Activity Monitoring. The overall structure is good and the novelty of the paper is clear. All figures and results are well written and discussed. The test validation is good. It would be good to see more comparisons and analysis. The overall paper is good.

Reviewer 5 Report

In this paper, the authors report a sensing device combining a single 6-axis IMU with a beam structure. A fusion algorithm is presented which combines IMU data in the wavelet domain to reduce intrinsic sensor noise.

As a reviewer, I think that the subject is consistent with the scope of journal. I would suggest the authors revised their manuscript to provide clear justifications for the following concerns.

1) The pictures(fig.5, fig.6) in the manuscript are with small words and of low resolution.

2) The manuscript lacks a comparison with current SOTA methods.

3) The authors should add some ablation experiments. whether there is temperature, filters or other parameters affects the results.

4) I think that the latest work on the topic in recent two years should be reviewed and summarized. Here are some suggestions for authors to go through and follow the direction for reworking. e.g.,

Liu, J., Zhang, X., Zhu, S., Li, X., Zhao, H., Li, X., ... & Miao, Y. (2023). Development and measurement of airborne oxygen sensor for aircraft inerting system based on TDLAS with pressure compensation and 2f/1f normalized method. Infrared Physics & Technology, 131, 104689.

5) At present, data and codes open source is the general trend. I suggest that the authors open the codes when the manuscript is modified, so that the academic communication can be better and the proposed method can stand the test of time.

Thank you

Round 2

Reviewer 1 Report

My comments have been addressed and the article can be considered for further editorial processing as per the journal's rules and regulations. Thank you

Reviewer 2 Report

Comments are addressed. it can be accepted

Reviewer 3 Report

the paper seems now ready

no issue